# Takin-VC: Zero-shot Voice Conversion via Jointly Hybrid Content and Memory-Augmented Context-Aware Timbre Modeling

## Abstract

Zero-shot voice conversion (VC) aims to transform the source speaker timbre into an arbitrary unseen one without altering the original speech content. While recent advancements in zero-shot VC methods have shown remarkable progress, there still remains considerable potential for improvement in terms of improving speaker similarity and speech naturalness. In this paper, we propose *Takin-VC*, a novel zero-shot VC framework based on jointly hybrid content and memory-augmented context-aware timbre modeling to tackle this challenge. Specifically, an effective hybrid content encoder, guided by neural codec training, that leverages quantized features from pre-trained HybridFormer and WavLM is first presented to extract the linguistic content of the source speech. Subsequently, we introduce an advanced cross-attention-based context-aware timbre modeling approach that learns the fine-grained, semantically associated target timbre features. To further enhance both speaker similarity and real-time performance, we utilize a conditional flow matching model to reconstruct the Mel-spectrogram of the source speech. Additionally, we advocate an efficient memory-augmented module designed to generate high-quality conditional target inputs for the flow matching process, thereby improving the overall performance of the proposed system. Experimental results demonstrate that the proposed Takin-VC method surpasses state-of-the-art zero-shot VC systems, delivering superior performance in terms of both speech naturalness and speaker similarity.

## 1 Introduction

Zero-shot voice conversion (VC) refers to the task of modifying the timbre of a source speech to match that of a previously unseen speaker, while preserving the original phonetic or linguistic content. This technology has found broad applications in various practical domains Gan et al. (2022); Tomashenko et al. (2022); Liu et al. (2021).

In recent times, zero-shot VC has witnessed great progressions, with numerous state-of-the-art (SOTA) approaches (Li et al., 2023a; Hussain et al., 2023; Choi et al., 2023; Anastassiou et al., 2024; Li et al., 2024; Luo & Dixon, 2024) exhibiting impressive results in converting natural and realistic utterances. The key idea behind these methods is to factorize speech into distinct elements, such as linguistic content and timbre elements, and then leverage the source speech content alongside the target speaker timbre to synthesize the desired target speech. In this paradigm, the quality of content and timbre representations, as well as the quality of their disentanglement, significantly impact their final performance. Consequently, numerous studies have sought to improve VC performance by designing more advanced modules (Wu et al., 2020; Wu & Lee, 2020; Tang et al., 2022; Wang et al., 2021; Yang et al., 2022a; Huang et al., 2023), information disentanglement approaches (Zhao et al., 2022; Tang et al., 2022; Dang et al., 2022; Yao et al., 2024b) and so forth. However, achieving high-quality decoupling of utterances into distinct components remains a challenging task (Pan et al., 2023; 2024a;c; Yao et al., 2024a), and existing zero-shot VC systems still exhibit subpar performance when handling unseen speakers primarily due to the underlying issues. First, current methods cannot fully mitigate the influence of source speaker timbre during the extraction of linguistic content features, a problem commonly referred to as "timbre leakage." Second, they normally employ pre-trained speaker-verification (SV) models to capture target timbre features and cast them

as globally time-invariant representations. However, as highlighted in (Jiang et al., 2024), the timbre representations may vary with the linguistic content, rendering the performance of these approaches less optimal. Recently, the advances in large-scale speech language models (Wang et al., 2023c; Borsos et al., 2023) have tried to tackle this issue by leveraging robust in-context learning capabilities for predicting target speech from concise utterances as prompts. Nevertheless, these methods may suffer from stability issues and error accumulation due to their auto-regressive nature, which can gradually degrade conversion quality.

To address the aforementioned limitations, we introduce *Takin-VC*, an effective VC framework with advanced modeling of content, timbre and audio quality in a zero-shot fashion. Specifically, we propose a hybrid content encoder guided by neural codec training that integrates the phonetic posteriorgrams (PPGs) features and quantized self-supervised learning (SSL) representations from two pretrained models, i.e., HybridFormer (Yang et al., 2023b) and WavLM (Chen et al., 2022), so as to precisely capture the linguistic content. For speaker timbre modeling, we first propose a content-aware timbre modeling method that employs cross-attention (CA) to leverage the target voiceprint (VP) features extracted from a pre-trained speaker verification (SV) model (Wang et al., 2023b), with the captured source content. This integration enables our proposed approach to learn target timbre representations associated with source content. Additionally, to further enhance speaker similarity, we advocate a memory-augmented module capable of generating high-quality conditional target inputs for a conditional flow matching (CFM) model Tong et al. (2023b), ultimately culminating in the synthesis of the target speech using a pre-trained vocoder (Lee et al., 2022).

To evaluate the performance of the Takin-VC system, we conduct extensive experiments on the both large-scale 500k-hour multilingual (Mandarin and English) and publicly available LibriTTS Zen et al. (2019) datasets. Experimental results demonstrate that Takin-VC consistently outperforms state-of-the-art (SOTA) zero-shot VC methods in terms of both speaker similarity and speech naturalness. Notably, Takin-VC achieves significant improvements in both subjective and objective metrics compared with all baseline systems, further validating its effectiveness. For more detailed speech samples, please visit our **demo page** [1].

In summary, the main contributions of this work are outlined as follows:

- We present Takin-VC, a robust and effective zero-shot VC framework that integrates advanced modeling capabilities for content, timbre, and speech quality. Takin-VC is capable of generating semantically coherent target timbre representations for unseen speakers, resulting in improved speaker similarity and enhanced naturalness/intelligibility.

- We introduce a hybrid linguistic content encoder that leverages the PPGs and quantized SSL features from the pre-trained HybridFormer and WavLM, with the guidance of neural codec-based training.

- We propose a context-aware timbre modeling approach based on CA to effectively integrate the source content and target timbre features, bridging the speaker similarity gap between the target speech and ground truth recording.

- We advocate a memory-augmented module to generate high-quality conditional target inputs for the CFM model, further boosting the speaker similarity performance of our proposed method.

## 2 BACKGROUND

**Zero-shot Voice Conversion.**

In contrast to previous few-shot Wang et al. (2020); Gabryś et al. (2022) and one-shot Tang et al. (2022); Li et al. (2023b) VC approaches, zero-shot VC presents a more challenging task, as it requires the model to generalize to unseen speakers without any additional training or fine-tuning. In recent years, advancements in deep learning techniques, such as SSL speech models and diffusion models, have led to significant progress in zero-shot VC. SEF-VC Li et al. (2024) learns speaker timbre from reference speech using a CA mechanism and reconstructs waveforms from HuBERT Hsu et al. (2021) tokens. Choi et al. (2023) introduced Diff-HierVC, a diffusion-based hierarchical VC

---

[1] https://anonymous.4open.science/w/takin-vc-0CD8/

method that uses XLS-R Babu et al. (2021) for content extraction and employs two diffusion models to generate high-fidelity converted pitch and Mel-spectrograms. The utilization of robust SSL features, which are rich in phonetic and paralinguistic nuances, has led to improved performance in these methods compared to prior works Fang et al. (2018); Kaneko et al. (2019). Despite these impressive results, SSL-based zero-shot VC approaches Dang et al. (2022); Hussain et al. (2023); Li et al. (2023a) may still encounter the timbre leakage problem, as SSL features do not explicitly disentangle timbre information, while diffusion-based VC methods Popov et al. (2021); Choi et al. (2024) often struggle with poor real-time performance. Another cutting-edge zero-shot VC paradigm Zhang et al. (2023); Wang et al. (2023c); Baade et al. (2024) involves decoupling speech into semantic and acoustic tokens using neural codecs (Défossez et al., 2022; Yang et al., 2023a; Pan et al., 2024b) and SSL speech models Chen et al. (2022); Baevski et al. (2020), subsequently leveraging language models to generate the converted speech. However, these methods still possess great potential for improvement regarding speaker similarity and naturalness/intelligibility.

**Flow Matching-based Generative Models.**

Recently, flow matching-based generative models Lipman et al. (2022); Tong et al. (2023c;a) have garnered considerable attention in the realm of generative tasks, particularly in the image generation task Ho et al. (2020); Saharia et al. (2022); Ruiz et al. (2023). These methods focus on approximating the transport probability path from random noise to the target distribution by estimating the associated vector field. By employing a neural ordinary differential equation (ODE), these models learn the optimal transport trajectory and establish a direct link between noise and target samples, which greatly reduces the required number of sampling steps. In contrast to diffusion-based methods Bartosh et al. (2023); Zhou et al. (2023); Zheng et al. (2023), flow matching offers improved training stability and real-time performance.

Influenced by this wave of innovation, the speech processing domain has begun to explore flow matching-based generative systems as well. For instance, SpeechFlow Liu et al. (2023) leverages a pre-trained generative model using flow matching and masked conditions on extensive untranscribed speech data, enabling effective adaptation to various downstream tasks like speech enhancement, separation, and so forth. ELaTE Kanda et al. (2024) is a zero-shot TTS system that generates natural laughter by mimicking voice characteristics from an audio prompt and precisely controlling laughter timing and expression through specific input cues. P-Flow Kim et al. (2024) utilizes speech prompts for speaker adaptation, integrating a speech-prompted text encoder that generates speaker-conditional representations with a flow matching generative decoder to achieve high-quality, real-time speech synthesis. Nevertheless, the application of flow matching in zero-shot voice conversion (VC) tasks is still in its developmental phase, indicating the urgent need for a stable and efficient flow matching-based zero-shot VC framework.

## 3 METHODS

### 3.1 OVERIVEW

As shown in Fig. 1, our Takin-VC system comprises four components: the hybrid linguistic content encoder, memory-augmented & context-aware timbre modeling approach, and CFM model.

To elaborate, the hybrid content encoder is designed to precisely capture linguistic content $x_{s_{cont}}$ by leveraging the complementary strengths of PPG and SSL features with the guidance of neural codec-based training. For timbre modeling, we extract Mel-spectrograms from randomly segmented reference waveform from the same speaker as the source speech, focusing on learning semantically correlated target timbre features and conditional target inputs for the CFM model, denoted as $x_{s_c t_t}$ and $x_{t_{cond}}$. In our case, the duration of the reference wav is $4s$. This process comprises two main components: context-aware timbre modeling and memory-augmented timbre modeling. The former begins by extracting the target VP features using a pre-trained speaker verification model[2]. These VP features are then concatenated with the reference Mel-spectrograms to form the key and value in the CA mechanism, while the attention query is derived from $x_{s_{cont}}$. The latter incorporates a stack of convolution, activation, and self-attention layers to generate high-quality conditional target

---

[2]https://modelscope.cn/models/iic/speech_campplus_sv_zh_en_16k-common_advanced

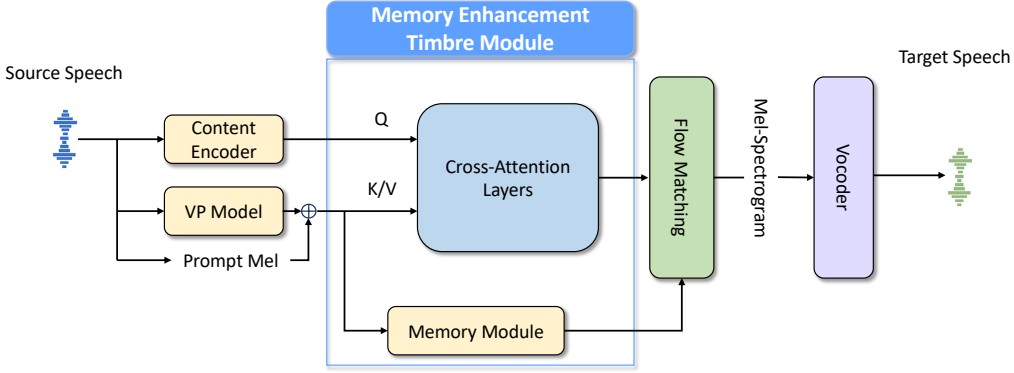

Figure 1: Overview of Takin VC.

inputs $x_{t_{cond}}$ for the CFM model, which will be detailed below. Finally, we employ a CFM model to reconstruct the source Mel-spectrograms based on $x_{s_c t_t}$ and $x_{t_{cond}}$, followed by employing a pre-trained Bigvgan vocoder to synthesize the desired target utterance.

## 3.2 HYBRID LINGUISTIC CONTENT ENCODER

Current zero-shot VC methods typically rely on pre-trained automatic speech recognition (ASR) Gulati et al. (2020); Yang et al. (2022b); Kim et al. (2022) methods or SSL speech models to extract linguistic content from the original waveform. Nonetheless, both approaches exhibit their respective limitations: the PPGs lacks certain essential paralinguistic nuances, while the latter does not explicitly disentangle timbre information. Therefore, to capture content representations with higher quality, our Takin-VC combines their merits through a neural codec training guided hybrid linguistic content encoder, as shown in Fig. 2.

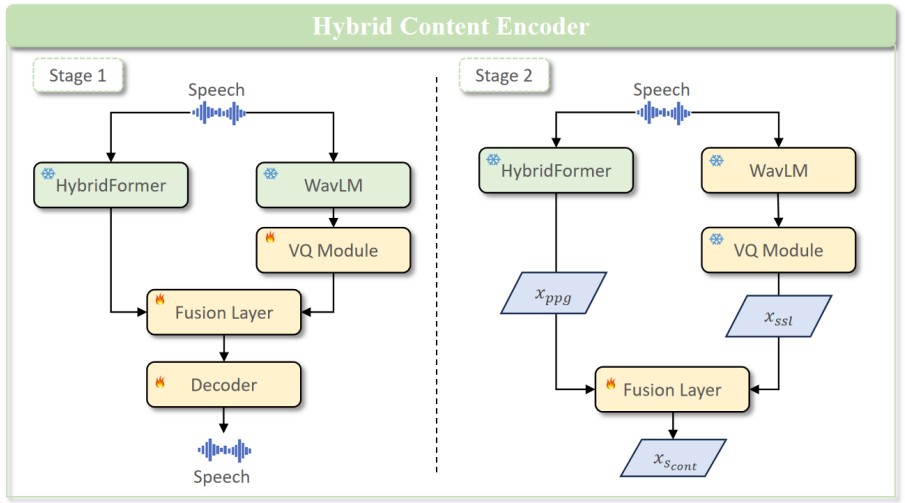

Figure 2: Content Encoder of Takin VC.

Formally, given an input source speech $x$, the proposed hybrid content encoder first extracts its corresponding PPG and SSL features, denoted as $x_{ppg}$ and $x_{ssl}$, using pre-trained HybridFormer and WavLM, respectively. For our scenario, the HybridFormer is trained on an in-house multilingual corpus of Mandarin and English, while the sixth-layer output features of WavLM are selected as $x_{ssl}$ for further processing.

However, merely relying on the combination of SSL and PPG features is insufficient to achieve optimal VC performance. To further enhance overall performance and address potential timbre leakage, we incorporate a neural codec-based training approach Pan et al. (2024b) for end-to-end training of the proposed hybrid content encoder, as depicted in the left part of Fig. 2. Concretely, we regard WavLM as the encoder in our neural codec framework and employ a residual vector quantization-based quantizer like Défossez et al. (2022) to discretize the SSL features. To effectively leverage the PPGs alongside the quantized SSL features, we introduce a simple yet effective fusion module designed to adaptively combine these elements. This module comprises Conv1D layers and ReLU-based gating mechanisms to integrate the SSL and PPG features. The fusion process is formulated as follows:

$$x_{s_{cont}} = \text{ReLU}^{\diamond}\left(\text{Conv1d}\left(\alpha_{ssl} \cdot VQ(x_{ssl}) + \alpha_{ppg} \cdot x_{ppg}\right)\right) \tag{1}$$

where $\alpha_{ssl}$ and $\alpha_{ppg}$ are learnable hyperparameters, and VQ denotes the vector quantization operation, while Conv1d and $\text{ReLU}^{\diamond}(*)$ represent the convolution and ReLU operations, respectively.

As a consequence, with the guidance of neural codec training, the quality of the fused hybrid SSL and PPG features can be significantly enhanced, resulting in improved naturalness and intelligibility in Takin-VC.

### 3.3 CONTEXT-AWARE & MEMORY-AUGMENTED TIMBRE MODELING

#### 3.3.1 CONTEXT-AWARE TIMBRE MODELING VIA CROSS-ATTENTION

Current mainstream VC methods typically regard speaker timbre as a global time-invariant representation (Lin et al., 2021; Li et al., 2024). Nevertheless, recent work (Jiang et al., 2024) has uncovered a close correlation between timbre modeling and content information.

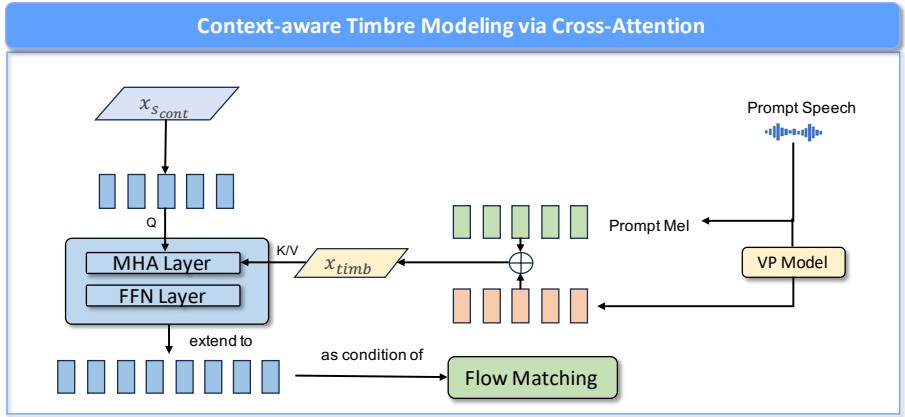

Figure 3: The structure of context aware timbre modeling in Takin VC.

Hence, drawing inspiration from this insight, we propose an innovative context-aware timbre modeling approach based on CA. First, we employ a pre-trained SV model to extract a target speaker's timbre embedding rather than using a global timbre encoder, and then concatenate it with the shuffled Mel-spectrograms of the target speech, denoted as $x_{t_{timb}}$, to minimize the influence of the target content. Subsequently, to learn semantically correlated timbre features that associate the source content with the timbre of the target speaker, we introduce an effective CA-based module. This module takes source content $x_{s_{cont}}$ as the query and $x_{t_{timb}}$ as both the key and value, consisting of a series of linear projection, multi-head CA, layer normalization, and position-wise feed-forward networks (FFN), as detailed in Fig. 3. Finally, we perform interpolation on the extracted features $x_{s_c t_t}$ to ensure that their dimensionality corresponds to that of the source Mel-spectrogram, thereby facilitating the subsequent training of the CFM model.

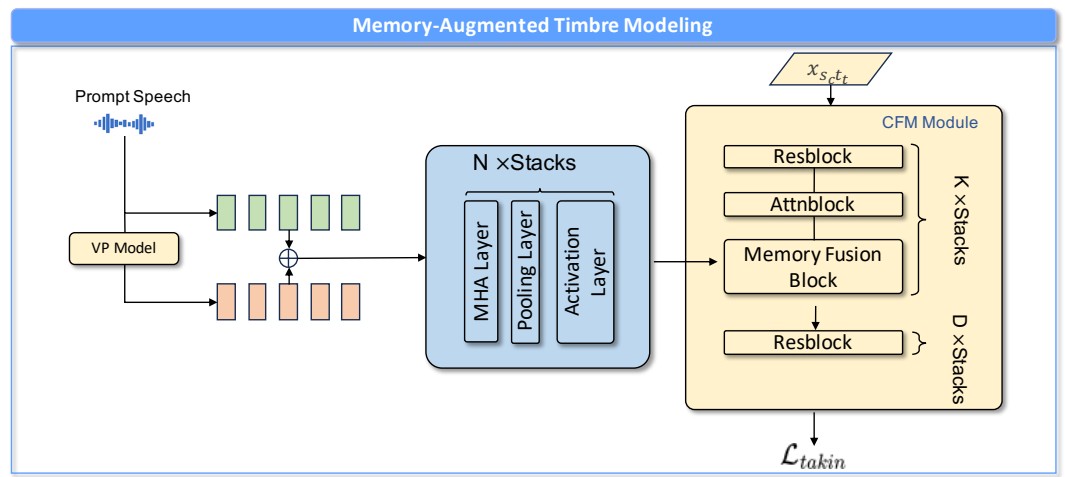

Figure 4: The structure of memory-augmented timbre modeling in Takin VC.

### 3.3.2 MEMORY-AUGMENTED TIMBRE MODELING

Since we use a CFM model to reconstruct the source Mel-spectrograms, obtaining high-quality conditional target inputs is quite essential, as they provide key guidance for training the CFM model. To this end, we design an efficient memory-augmented module that adaptively integrates the Mel-spectrogram and VP features of the reference speech, as outlined in Fig. 4. To be specific, our proposed memory-augmented module initially use a Conv1d layer to project the $x_{ref}$ to a latent feature space. Subsequently, we incorporate multiple SA blocks, each containing several group normalization, multi-head SA, and 1D Conv layers, followed by a Conv1d layer and a LeakyReLU activation layer. This design effectively leverages these features in a stable and learnable manner. At the end of memory module, we compute the average vector of the obtained representations across the time dimension to produce the final output $x_{t_{cond}}$. Finally, $x_{t_{cond}}$ is input into the Memory Fusion Layer (a combination of the Gated Activation Layer and FiLM Layer Perez et al. (2018)) within the flow matching network to reconstruct the Mel-spectrogram.

### 3.4 CONDITIONAL FLOW MATCHING-BASED DECODER

In Takin-VC, to facilitate more efficient training and faster inference, we leverage a CFM model with optimal-transport (OT-CFM) to approximate the distribution of source Mel-spectrograms and generate outputs conditioned on $x_{s_{ctt}}$ and $x_{t_{cond}}$, all in a simulation-free manner.

Assume that the standard distribution and target distribution are denoted as $p_0(x)$ and $p_1(x)$, respectively. The OT flow $\phi : [0,1] \times R^d \to R^d$ establishes the mapping between two density functions through the use of an ordinary differential equation (ODE):

$$\frac{d}{d_t}\phi_t(x) = v_t(\phi_t(x), t)$$

$$\phi_0(x) \sim p_0(x) = \mathcal{N}(x; 0, I), \quad \phi_1(x) \sim p_1(x) \tag{2}$$

where $v_t$ is a learnable time-dependent vector field, and $t \in [0,1]$. Since multiple flows can generate this probability path, making it challenging to determine the optimal marginal flow, we adopt a simplified formulation, as proposed in Tong et al. (2023b):

$$\phi_{t,z}^{OT}(x) = \mu_t(z) + \sigma_t(z)x$$

$$\mu_t(z) = (1-(1-\sigma_{min})t)z, \quad \sigma_t(z) = t \tag{3}$$

where $z$ represents the random variable, $\sigma_{min}$ is a hyper-parameter set to 0.0001. As a consequence, the final training objective of the proposed Takin-VC can be formulated as:

$$\mathcal{L}_{takin} = \mathbb{E}_{t,p(x_0),q(x_1)} \|((x_1 - (1 - \sigma)x_0) - v_t(\phi_{t,x_1}^{OT}(x_0)|\theta, h)\|^2 \qquad (4)$$

where $\theta$ is the weights of the flow matching model, $h$ is the conditional input $x_{t_{cond}}$.

# 4 EXPERIMENTAL SETUP

## 4.1 BASELINE SYSTEM

We conduct a comparative experiment of the performance in zero-shot voice conversion between our proposed Takin-VC approach and baseline systems, encompassing the following system:

- DiffVC (Popov et al., 2021): A zero-shot VC system based on diffusion probabilistic modeling, which employs an averaged mel spectrogram aligned with phoneme to disentangle linguistic content and timbre information.

- NS2VC[3]: A modified voice conversion edition of NaturalSpeech2 (Shen et al., 2023), which employ both diffusion and codec model to achieve zero-shot VC.

- VALLE-VC (Wang et al., 2023a): We replace the original phoneme input with the semantic token extracted from the supervised model to make VALLE convert the timbre of source speech to the target speaker.

- SEFVC (Li et al., 2024): A speaker embedding free voice conversion model, which is designed to learn and incorporate speaker timbre from reference speech via a powerful position-agnostic CA mechanism and then reconstruct waveform from HuBERT semantic tokens in a non-autoregressive manner.

## 4.2 EVALUATION METRICS

To evaluate the performance of our proposed Takin-VC and baseline systems, both subjective and objective metrics are introduced. For subjective metrics, we employ naturalness mean opinion score (NMOS) to evaluate the naturalness of the generated samples and similarity mean opinion scores (SMOS) to evaluate the speaker similarity. We invite 20 professional participants to listen to the generated samples and provide their subjective perception scores on a 5-point scale: '5' for excellent, '4' for good, '3' for fair, '2' for poor, and '1' for bad. For objective metrics, we employ word error rate (WER), UTMOS, and speaker embedding cosine similarity (SECS) to evaluate the intelligibility, quality, and speaker similarity. Specifically: 1) We use a pre-trained CTC-based ASR model[4] to transcribe the generated speech and compare with ground-truth transcription; 2) We use a MOS prediction system that ranked first in the VoiceMOS Challenge 2022[5] to estimate the speech quality of the generated samples; 3) We use the WavLM-TDCNN speaker verification model[6] to measure speaker similarity between generated speech and target speech.

## 4.3 DATASET

### 4.3.1 SMALL SCALE DATASET

We employ the LibriTTS dataset to train our system and baseline systems, which contain 585 hours of recordings from 2,456 English speakers. We follow the official data split, using all training datasets for model training and "dev-clean" for model selection. The "test-clean" dataset is used to construct the evaluation set. All samples are processed at a 16kHz sampling rate.

---

[3]`https://github.com/adelacvg/NS2VC`
[4]`https://huggingface.co/facebook/hubert-large-ls960-ft`
[5]`https://github.com/tarepan/SpeechMOS`
[6]`https://github.com/microsoft/UniSpeech/tree/main/downstreams/speaker_`
`verification`

### 4.3.2 Large Scale Dataset

To train a robust Takin VC model, we collected a dataset of approximately 500k hours. During the data collection process, we used an internally constructed data pipeline specifically designed for audio large model tasks. This pipeline includes signal-to-noise ratio (SNR) filtering, audio spectrum filtering (filtering out 24k audio with insufficient high-frequency information and pseudo 24k audio), VAD (Voice Activity Detection), LiD+ASR (Language Identification + Automatic Speech Recognition), speaker separation and identification, punctuation prediction, and background noise filtering. Regarding the test set, to validate the effectiveness of the Takin-VC model, we collected speech data from the internet that includes 100 non-preset speakers for evaluation. These speakers represent a variety of attributes such as gender, age, language, and emotion to ensure a comprehensive evaluation of the model's performance.

### 4.4 Model Configuration

For the content encoder part, in the first stage, we used the 12-layer HybridFormer-base model trained on a large dataset of 500K hours. For the wavlm part, we used the output features of the 6th layer. In the VQ part, we adopted a single-layer 8200 codebook with a hidden dimension of 1024, trained for 1 million steps on 100K hours of data. The fusion layer, as described in Sec. 3.2, is a simple process of conv1d, activation layer, and weighted summation. The Decoder adopts the same structure and configuration as Hificodec Yang et al. (2023a).

In the part of timbre modeling and flow matching restoration, both the context-aware timbre module and the memory module use a transformer layer with 8 heads, 6 layers, and a hidden size of 1024, with only the form of attention being different. The main structure of flow matching uses a design of 10-layer Unet plus 3 layers of resblock, with a hidden size of 1280. A Memory Fusion Block is inserted into the 10-layer Unet to enhance the timbral similarity of the generated audio.

For the small-data experiments, we used four A800 GPUs, whereas the large-data experiments were conducted on eight A800 servers. The batch size on each card was set to 16, and the AdamW learning rate was set to 1e-4. In the inference section, experiments typically took 15 to 50 steps, with the final table uniformly adopting the results of 50 steps. The Classifier-Free Guidance (CFG) coefficient ranged from 0.1 to 1.0, with 0.7 used in the table. The specific experimental results will be detailed later.

## 5 Experimental Results

### 5.1 Experiments on small dataset

We first evaluate the performance of our proposed Takin-VC using subjective metrics. These metrics capture human perception of the enhanced speech's naturalness, intelligibility, and speaker similarity. As shown in Table 1, we can find that 1) our proposed system achieves the highest NMOS of 3.98, which is significantly higher than baseline systems; 2) the speaker similarity of our proposed system also outperforms all baseline systems. These results demonstrate that Takin-VC can achieve superior performance than the baseline system in the perceived aspect.

Table 1: Comparison results of subjective and objective metrics between Takin-VC and the baseline systems in zero-shot voice conversion. Subjective metrics are computed with 95% confidence intervals and "GT" refers to ground truth samples.

|  | NMOS (↑) | SMOS (↑) | WER (↓) | UTMOS (↑) | SECS (↑) |
|---|---|---|---|---|---|
| GT | 4.17±0.04 | - | 2.04 | 4.21 | - |
| DiffVC | 3.75±0.05 | 3.66±0.07 | 3.08 | 3.68 | 0.61 |
| NS2VC | 3.65±0.07 | 3.51±0.06 | 2.94 | 3.64 | 0.53 |
| VALLE-VC | 3.80±0.06 | 3.79±0.04 | 2.77 | 3.72 | 0.65 |
| SEFVC | 3.68±0.05 | 3.76±0.06 | 3.75 | 3.51 | 0.63 |
| Takin-VC | **3.98**±0.04 | **4.11**±0.05 | **2.35** | **4.08** | **0.71** |

Furthermore, we evaluate the performance using objective metrics. The WER of our proposed system is 2.35, only slightly higher than the ground truth samples, indicating that the samples generated by Takin-VC exhibit better intelligibility. Moreover, Takin-VC achieves a UTMOS of 4.08 and an SECS of 0.71, demonstrating superior quality and similarity performance. Overall, the objective results of our proposed Takin-VC outperform all baseline systems and further corroborate the subjective findings.

## 5.2 EXPERIMENTS ON LARGE DATASET

We employ the large scale dataset to train our proposed Takin-VC and investigate the performance in different conversion scenarios across different gender. As shown in Table 2, we divide the experiments into four groups: female to female (F2F), female to male (F2M), male to male (M2M), and male to female (M2F) to investigate performance differences. The results show that all metrics outperform Takin-VC trained on a smaller dataset, demonstrating that our proposed approach scales effectively. Additionally, the conversion results for same-gender conversions are slightly better than cross-gender conversions in both SMOS and SECS, while other metrics remain similar across all four group settings.

Table 2: Detailed results of Takin-VC on different conversion scenarios. "F" and "M" represent the female and male, respectively.

|     | NMOS (↑) | SMOS (↑) | WER (↓) | UTMOS (↑) | SECS (↑) |
|-----|----------|----------|---------|-----------|----------|
| GT  | 4.21±0.05 | -       | 2.11    | 4.18      | -        |
| F2F | 4.16±0.04 | 4.18±0.03 | 2.11  | 4.11      | 0.74     |
| F2M | 4.14±0.05 | 4.09±0.05 | 2.24  | 4.13      | 0.71     |
| M2M | 4.12±0.04 | 4.11±0.04 | 2.20  | 4.20      | 0.73     |
| M2F | 4.13±0.05 | 4.04±0.06 | 2.31  | 4.09      | 0.70     |

To further investigate the speaker similarity performance of our Takin-VC, we use the t-SNE method (Van der Maaten & Hinton, 2008) to visualize the speaker embeddings of 13 speakers, comparing the ground truth samples with the converted samples generated by Takin-VC. As shown in Figure 5, the embeddings of real and converted speech from the same speaker are closely clustered. This demonstrates that the speech generated by Takin-VC closely matches real human speech in both quality and speaker similarity.

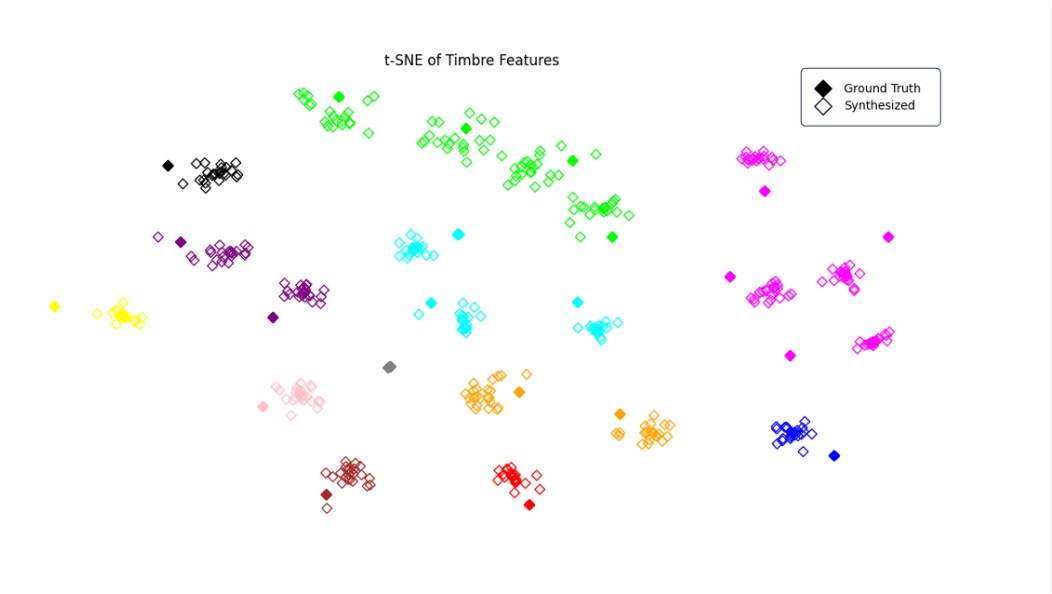

Figure 5: The t-SNE result of speaker similarity between ground truth samples and converted speech.

## 5.3 ABLATION STUDY

We conduct ablation experiments to evaluate the effectiveness of each component in our proposed system for generating natural-sounding samples and accurate timbre modeling. As shown in Table 3, NMOS and WER results degrade when we replace the proposed hybrid content encoder with a conventional ASR encoder. This suggests that the conventional ASR encoder is less capable of disentangling linguistic content from the necessary paralinguistic information, underscoring the importance and effectiveness of our hybrid encoder in extracting linguistic content. Additionally, we observe a notable decline in speaker similarity when the voice print is removed from the attention module. We believe the voice print introduces a stronger timbre bias, which helps the attention module focus on capturing timbre information. Furthermore, when we remove the memory module, SMOS and SECS scores show significant degradation compared to the original Takin-VC, demonstrating the critical role of the memory module in improving timbre modeling. These ablation results demonstrate the effectiveness of each component proposed in our Takin-VC.

Table 3: Experimental results on ablation studies. "w/o vp" represents removing voice print in the attention module. "w/o hybrid" represents replacing the proposed hybrid content encoder with the conventional used ASR encoder, and "w/o memory" means removing the timbre memory module.

|            | NMOS (↑)    | SMOS (↑)    | WER (↓) | UTMOS (↑) | SECS (↑) |
|------------|-------------|-------------|---------|-----------|----------|
| Takin-VC   | 3.98±0.04   | 4.11±0.05   | 2.35    | 4.08      | 0.71     |
| w/o hybrid | 3.67±0.04   | 4.01±0.04   | 2.79    | 3.89      | 0.66     |
| w/o vp     | 3.94±0.05   | 3.89±0.04   | 2.51    | 3.98      | 0.61     |
| w/o memory | 3.92±0.04   | 3.75±0.05   | 2.44    | 4.01      | 0.52     |

## 6 DISCUSSION AND LIMITATIONS

Takin is an effective and data-efficient zero-shot VC system that achieves comparable naturalness and speaker adaptation performance to its large-scale, autoregressive counterparts. The core of this approach lies in the neural codec training based hybrid linguistic content encoder, which captures high-quality speaker-agnostic content representations, and the introduction of both context-aware timbre modeling and memory-augmented modules to enhance speaker similarity performance. In many ways, our work provides a strong foundation for future studies, as we demonstrate that state-of-the-art performance in this task can be achieved without relying on complex training setups, representation quantization steps, or costly autoregressive models.

This work primarily focuses on zero-shot capabilities for speech generation, while zero-shot capabilities for speech editing remain limited and are a subject for future exploration. Additionally, while high-quality zero-shot VC has great potential, it can also lead to negative social impacts, such as voice impersonation of public figures and non-consenting individuals. We highlight this as a potential misuse of the technology to raise awareness of its ethical implications.

## 7 CONCLUSIONS

In this study, we propose a novel framework called Takin-VC, designed to achieve high quality and similarity in zero-shot VC. We introduce an effective neural codec training guided hybrid content encoder that leverages quantized features from both pre-trained HybridFormer and WavLM to extract the linguistic content of the source speech. This hybrid content encoder improves the naturalness and intelligibility of the converted speech. Additionally, we present an advanced cross-attention-based, context-aware timbre modeling approach that captures fine-grained, semantically associated target timbre features. Furthermore, we employ a conditional flow-matching model to efficiently reconstruct the Mel-spectrogram of the source speech and propose an efficient memory-augmented module for the flow-matching process, enhancing the overall performance of the generated samples. Experimental results demonstrate that Takin-VC outperforms all baseline systems in naturalness and speaker similarity on benchmark datasets. Ablation studies also confirm the effectiveness of each component in our approach.

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
