# OpenReview forum: "Takin-VC: Zero-shot Voice Conversion via Jointly Hybrid Content and Memory-Augmented Context-Aware Timbre Modeling"
_ICLR.cc/2025/Conference — ICLR 2025 Conference Withdrawn Submission_

### Official Review · Reviewer_R8GT · 2024-10-21

**Soundness:** 3
**Presentation:** 2
**Contribution:** 2
**Rating:** 3
**Confidence:** 3

**Summary:**

This paper proposes a zero-shot voice conversion method that enhances audio quality by addressing timbre leakage. The approach shows improvements over existing methods through extensive experiments. However, its novelty is limited, relying on established techniques. Additional clarity and further ablation studies are recommended.

**Strengths:**

This paper proposes a zero-shot voice conversion method that uses context-aware timbre features and a multi-stage pertained content encoder to solve the timbre leakage problem and enhance the naturalness and quality of the converted audio. They also include comprehensive experiments, including ablation studies and evaluations on small and large datasets to show improvements over the state-of-the-art (SOTA) in both subjective and objective metrics, which suggests that the method can improve both speaker similarity and the naturalness of converted speech.

**Weaknesses:**

1. While the integration of neural codec-based training, context-aware timbre modeling, and flow matching is interesting, the novelty of the contribution seems limited, with much of the work focusing on combining well-established techniques rather than introducing fundamental innovations.
2. The authors mention they improve real-time performance but lack further analysis and experiments to show how they improve it.
3. The figures can be improved to stand in path with the used notations and modules in this paper, or it can make the model architecture hard to understand.
4. The paper takes cross-attention between the content and timbre feature as one of the contributions. However, many existing works like GPT-SoVITS have already carried out methods similar to it.
5. Ablation studies should be carried out on the neural codec training.
6. The author should give more explanation on neural codec training like the objective function and when the neural codec training takes place (i.e. perform neural codec training first or together with other training processes.)

**Questions:**

1. How to prevent overfitting during neural codec training?
2. For the neural codec involved in Mel and FBank features besides the encoder, how to directly generate the speech from the fused features of WavLM and HybridFormer in Figure 2?

---

### Official Review · Reviewer_wkqP · 2024-11-01

**Soundness:** 3
**Presentation:** 2
**Contribution:** 2
**Rating:** 5
**Confidence:** 5

**Summary:**

The Takin-VC method is proposed, further enhancing the naturalness of the results while ensuring timbre conversion. An encoder is designed that integrates PPG and SSL features, compensating for the shortcomings of both types of features and better extracting semantic content. The extraction results of the Mel spectrogram and speaker features are fused as key-value pairs. The target timbre is integrated using a cross-attention mechanism. A memory-augmented module is utilized to generate conditions for the CFM, further improving timbre performance.

**Strengths:**

The effectiveness of the VC task is enhanced by combining some existing models or methods, with the most notable feature being the design of the encoder. The writing style is good, and the narrative is clear. The entire method is described with sufficient accuracy. The decoupling operations of timbre and semantic content prior to CFM hold certain reference significance in the current context of rapidly advancing large models.

**Weaknesses:**

The generation of key-value pairs utilizes reference Mel and VP features. It needs to be supplemented in the ablation experiments to determine whether the results would change if only VP features were used. Additionally, the paper mentions that SSL features do not clearly decouple timbre information. Currently, the authors refine semantic information through VQ operations, but many studies suggest that VQ does not guarantee complete decoupling of timbre and semantic content. It is recommended that the authors further test the effectiveness of VQ in their experiments.

**Questions:**

All my questions have been raised in the Weaknesses.

---

### Official Review · Reviewer_Jb2q · 2024-11-03

**Soundness:** 2
**Presentation:** 3
**Contribution:** 2
**Rating:** 3
**Confidence:** 4

**Summary:**

This work appears to be backed by substantial engineering efforts, enhancing zero-shot voice conversion performance through two main techniques: Jointly Hybrid Content and Memory-Augmented Context-Aware Timbre Modeling.

**Strengths:**

1. The authors provide a clear and thorough explanation of the Jointly Hybrid Content and Memory-Augmented Context-Aware Timbre Modeling techniques.
2. The proposed model achieves competitive results when compared to existing baselines.

**Weaknesses:**

1. The field of zero-shot voice conversion (VC) is somewhat niche, which may limit its broader impact. Moreover, with the impressive progress in related tasks such as zero-shot TTS (voice cloning), the potential for further development in zero-shot VC may be constrained.
2. The novelty of the approach is limited. The Jointly Hybrid Content module is essentially a combination of HybridFormer and WavLM representations, while Context-Aware Timbre Modeling via Cross-Attention is a common technique. Additionally, the Memory-Augmented Timbre Modeling component is a relatively simple block and lacks significant theoretical contributions.
3. Differences in the training dataset could introduce data bias.
4. It would be beneficial to compare the proposed model against more recent baselines, such as CosyVoice-VC and Seed-VC. Although I understand that comparing with works from the last few months might be unfair, demonstrating that Takin-VC significantly outperforms contemporaneous methods would showcase its potential.
5. The ablation studies do not demonstrate notable improvements in terms of audio quality or robustness.

**Questions:**

nothing.

---

### Official Review · Reviewer_wfZ4 · 2024-11-03

**Soundness:** 3
**Presentation:** 2
**Contribution:** 2
**Rating:** 3
**Confidence:** 4

**Summary:**

This paper proposed TAKIN-VC to deal with zero-shot TTS by hybrid content encoding and memory-augmented context-aware timbre modeling. The experimental outcomes presented demonstrate that TAKIN-VC outperforms existing state-of-the-art voice conversion (VC) systems, highlighting its effectiveness and potential in enhancing speech synthesis technology.

**Strengths:**

1. Figure 5 effectively illustrates the speaker similarity between the ground truth and the converted speech, providing clear visual evidence of the model's capability to maintain speaker characteristics.
2. Both subjective and objective evaluations demonstrate the effectiveness of the proposed method in achieving high-quality voice conversion. These results provide a comprehensive understanding of the model's performance from multiple perspectives.

**Weaknesses:**

1. Given that the output of the task is audio, it would be beneficial to provide some demonstrations either through a demo page or supplementary materials to better showcase the audio results.
2. Some notations in Sec. 3.1 is not easy to understand and remember, which might hinder the clarity and accessibility of the paper.
3. In the hybrid content encoder, SSL features are merely quantized, raising concerns about whether the timbre information contained could still potentially leak. The paper notes that PPG features can offer disentangled timbre information for SSL features; therefore, a comparative analysis regarding whether to use PPG or the use of PPG versus the use of more purified semantic information (such as text) is expected to clarify the effectiveness and security of the feature handling. The results of the ablation study can only demonstrate that using PPG is not enough.
4. The context-aware timbre modeling seems important, while the analysis is relatively inadequate. It would be insightful to investigate what the performance implications would be if only a global timbre embedding from the VP model were used, thereby understanding the significance of local context in timbre adaptation more deeply.
5. It would be better to explain more about the large-scale dataset and show some examples of this dataset.
6. The input of the memory module is x_sctt in Figure 4, but according to the description in Sec. 3.2.2, the input of the memory module should be x_ref.

**Questions:**

See the above weaknesses.

---

### Note · Authors · 2024-11-25

I have read and agree with the venue's withdrawal policy on behalf of myself and my co-authors.